# Design of Hybrid Polymeric-Lipid Nanoparticles Using Curcumin as a Model: Preparation, Characterization, and In Vitro Evaluation of Demethoxycurcumin and Bisdemethoxycurcumin-Loaded Nanoparticles

**DOI:** 10.3390/polym13234207

**Published:** 2021-11-30

**Authors:** Krissia Wilhelm Romero, María Isabel Quirós, Felipe Vargas Huertas, José Roberto Vega-Baudrit, Mirtha Navarro-Hoyos, Andrea Mariela Araya-Sibaja

**Affiliations:** 1Laboratorio BIODESS, Escuela de Química, Universidad de Costa Rica, San Pedro de Montes de Oca, San José 2060, Costa Rica; krissia.wilhelm@ucr.ac.cr (K.W.R.); maria.quirosfallas@ucr.ac.cr (M.I.Q.); luis.vargashuertas@ucr.ac.cr (F.V.H.); mnavarro@codeti.org (M.N.-H.); 2Laboratorio Nacional de Nanotecnología LANOTEC-CeNAT-CONARE, Pavas, San José 1174-1200, Costa Rica; jvegab@gmail.com; 3Laboratorio de Investigación y Tecnología de Polímeros POLIUNA, Escuela de Química, Universidad Nacional de Costa Rica, Heredia 86-3000, Costa Rica

**Keywords:** drug delivery, polymer-lipid hybrid nanoparticles, curcumin, demethoxycurcumin, bisdemethoxycurcumin, Pluronic F-127

## Abstract

Polymeric lipid hybrid nanoparticles (PLHNs) are the new generation of drug delivery systems that has emerged as a combination of a polymeric core and lipid shell. We designed and optimized a simple method for the preparation of Pluronic F-127-based PLHNs able to load separately demethoxycurcumin (DMC) and bisdemethoycurcumin (BDM). CUR was used as a model compound due to its greater availability from turmeric and its structure similarity with DMC and BDM. The developed method produced DMC and BDM-loaded PLHNs with a size average of 75.55 ± 0.51 and 15.13 ± 0.014 nm for DMC and BDM, respectively. An FT-IR analysis confirmed the encapsulation and TEM images showed their spherical shape. Both formulations achieved an encapsulation efficiency ≥ 92% and an exhibited significantly increased release from the PLHN compared with free compounds in water. The antioxidant activity was enhanced as well, in agreement with the improvement in water dissolution; obtaining IC_50_ values of 12.74 ± 0.09 and 16.03 ± 0.55 for DMC and BDM-loaded PLHNs, respectively, while free curcuminoids exhibited considerably lower antioxidant values in an aqueous solution. Hence, the optimized PHLN synthesis method using CUR as a model and then successfully applied to obtain DMC and BDM-loaded PLHNs can be extended to curcuminoids and molecules with a similar backbone structure to improve their bioactivities.

## 1. Introduction

In the last years, polymeric nanoparticles (PNs) have been one of the most studied nanocarriers to achieve drug delivery challenges [1]. This nanosystem possess high structural integrity afforded by the rigidity of the polymer matrix [2]; therefore, presenting advantages such as simple preparation and design, good biocompatibility, broad-structure variety, and notable bio-imitative properties [3]. However, the main drawback of PN is their low encapsulation efficiency of water-soluble drugs due to the fast leakage of the drug from the nanoparticles during the high-energy emulsification step commonly employed in their preparation [4,5]. In this scenario, the new generation of PN, the polymeric-lipid hybrid nanoparticles (PLHNs) emerged as a combination of PN with lipid-derived nanoparticles. It possesses the characteristics and the advantages of both polymer and lipid-based particles [5,6] and because of its hydrophobic matrix and hydrophilic core, it is possible to load both lipid soluble and water-soluble drugs within the same particle [5]. PLHNs consist of a central polymer core that is surrounded by single or multiple lipids [7], in which the therapeutic substances are encapsulated, an inner lipid layer enveloping the polymer core, whose main function is to confer biocompatibility to the polymer core, and an outer lipid-polymeric layer [5,6]. The outer coating acts not only as a barrier toward diffusion but also prolongs the in vivo circulation time of PLHNs in systemic circulation [5]. In addition, the inner lipid layer slows down the polymer degradation rate, and also maintains functions as a molecular fence minimizing leakage of the encapsulated content during the preparation and increasing drug loading efficiency [8]. The typical structure representation of a PLHN is shown in Figure 1.

For drug delivery and PN preparation, poloxamers, a polyethylene oxide-polypropylene oxide-polyethylene oxide (PEO-PPO-PEO)-based triblock, commercialized under the Pluronics^®^ trademark is one of the most used copolymers [9,10]. Its monomers can render an amphiphilic character in an aqueous solution based on the PEO solubility in water and the PPO insolubility. The PEO blocks are thus hydrophilic, while the PPO block is hydrophobic [11,12]. Besides leading to surface active properties, the block segregation also gives rise to self-assembly useful nanostructures [11]. Pluronic F-127 (PEO_100_–PPO_65_–PEO_100_) is relatively non-toxic to cells and widely investigated because its great hydrophobic region favors micellization [13]. It is the preferred one among pluronics for drug delivery applications due its high biocompatibility [14] and is approved by FDA as excipient in oral, ophthalmic, and topical medicinal formulations [15]. In terms of formulation of nanoparticles, Pluronic 127 possesses different functional attributes that subsequently influence the core-shell lipid-polymer nanoparticulate system and directly affect the efficiency of the delivery system [9]. Recent studies have shown the positive influence of Pluronic F-127 on the loading enhancement of curcumin [15,16].

Turmeric has a great variety of curcuminoids which has generated interest in the last years due to their recognized bioactivities including antioxidant, anti-inflammatory, anti-microbial, anti-diabetic, and immunomodulatory [17,18,19]. Among them, bis-demethoxycurcumin (BDM), demethoxycurcumin (DMC) and curcumin (CUR) are the most representative curcuminoids in turmeric. CUR is the major component making up 80%, DMC 17%, and BDM 3% in commercially available crude products [20,21]. Hence, CUR has been the most extensively studied curcuminoid and has been attributed a broad range of biological activities [22,23,24,25]. In addition, different types of nanosystems for drug delivery with several therapeutic applications have been applied to CUR [26,27,28,29], mainly in an effort to bypass the glucuronidation pathway [17] but also to improve its low solubility and chemical instability [30]. The chemical structures of the three curcuminoids are shown in Figure 1. The similarity in their structures suggests that they may exhibit similar bioactivities. Indeed, there are reports studying the anti-inflammatory and anticancer properties of DMC and BDM [31,32]. In turn, evidence indicates DMC and BDM possess the same drawbacks compromising CUR’s bioavailability [33,34]. Hence, these two curcuminoids are promising molecules for the improvement of their limited bioavailability through PLHNs.

In this study, we designed and optimized a simple method for the preparation of Pluronic F-127-based PLHNs able to load DMC and BDM individually using CUR as a model molecule because of its major presence in turmeric. Processing parameters including, ultrasonic probe, high-speed homogenization, mixing phases and homogenization speed were evaluated. Organic and aqueous phase’s composition: lipid, drug loading, lipid-drug ratio, organic solvent, surfactant, and polymer amount were investigated. The developed method was applied to obtain DMC and BDM-loaded PLHNs. In addition, the characterization of the nanoparticles using the FT-IR spectroscopy, DLS TEM and DSC, and the evaluation of their in vitro release and antioxidant activity, were performed.

## 2. Materials and Methods

### 2.1. Materials

Curcumin (CUR), demethoxycurcumin (DMC), and bisdemethoxycurcumin (BDM) were obtained and isolated from Curcuma longa by BIODESS Laboratory (Costa Rica). CUR, DMC, and BDM analytical standards used in the UHPLC and UV quantification studies as well as Cholesterol (Chol) poloxamer 407 (Pluronic F-127), 2,2-diphenyl-1-picrylhidrazyl (DPPH), dichloromethane (CH_2_Cl_2_), phosphoric acid (H_3_PO_4_) and disodium hydrogen phosphate were purchased from Sigma-Aldrich. Sodium dihydrogen phosphate monohydrate was acquired from Merck. Polysorbatum 80 (Tween 80) was purchased from Sonntag and Rote S.A., and sorbitan monooleate (Span^®^ 80) was supplied by LABQUIMAR S.A. Chloroform (CHCl_3_), methanol (MeOH), and acetonitrile (MeCN) were purchased from JTBaker. All solvents were HPLC/UV grade or highly pure, and the water was purified using a Millipore system filtered through a Millipore membrane 0.22 µm Millipak 40.

### 2.2. Design and Optimization of the Polymer-Lipid Hybrid Nanoparticles (PLHNs) Method Using CUR as Model

The strategy for the design of the PLHN was to combine one method for preparing solid lipid nanoparticles with another to prepare polymeric nanoparticles. Therefore, the emulsion method reported by Rompicharla et al. 2017 [35] and the emulsification solvent diffusion method reported by Udompornmongkol et al. 2015 [36], respectively, were used as starting methods. The following were the initial parameters: the aqueous solution composed of 5 mg/mL of Pluronic F-127 prepared in acetic acid 0.1% and Tween 80: Span80 1:1 4% was added dropwise into the organic one containing a CUR-lipid ratio of 1:24 in a 1:1 mixture of MeOH: CHCl_3_. Mixed phases were homogenized at 10,000 rpm for 10 min.

The optimization of the PLHN preparation method consisted of using different components of organic and aqueous phases as well as parameters related to mixing the phases and homogenization techniques for visually obtaining an emulsion and efficient formulations in terms of drug encapsulation. Hence, at each change, if the emulsion formation was visually confirmed, the next step was to evaluate the encapsulation efficiency (EE) both direct (EE_D_) and indirect (EE_I_) according to Equations (1) and (2), respectively, showed in the Section 2.3.1. Hence, the emulsion was placed in magnetic stirring for 10 min to eliminate the traces of the organic solvents. The particles were collected by ultracentrifugation and the CUR content was determined to calculate EE_I_. Then, the emulsion was washed three times with ultrapure water to remove the remaining unencapsulated molecule and unreacted substances. Further, EE_D_ was calculated by determining the CUR content in 100 μL of the final formulation dissolved in 900 μL of MeOH. A blank of PLHN was prepared using the same parameters tested without adding the CUR in the organic phase.

#### 2.2.1. Homogenization Technique

Two homogenization techniques were evaluated, first, a Cole Palmer Gex 30 Ultrasonic Processor (Cole Palmer, Illinois, IL, USA) operated at 130 watts and 20 KHertz provided with a 3 mm titanium probe. The aqueous and organic phases were mixed in a beaker in a one-time addition, and the solution was sonicated during two different periods of 30 and 45 min at intervals of 5 min. The second one consisted of a high speed ULTRA-TURRAX^®^ T25 homogenizer (IKA, Staufen, Germany). The aqueous solution was added into the organic one by testing three different speeds: dropwise (slow), approximately 3 mL/min (medium), and one-time addition (fast). Further, 10,000, 12,000 and 16,000 rpm homogenization speed during 10 min were tested.

#### 2.2.2. Lipid, Drug Loading, Lipid-Drug Ratio, and Organic Solvent

Chol, cocoa butter, cetyl palmitate, stearic acid, and a 1:1 mixture of Chol: cetyl palmitate were used. Once the lipid was selected, three different amounts of CUR 7.5, 10 and 15 mg as well as 12:1 and 48:1 drug-lipid ratios were tested. The organic phase composition was evaluated by testing 6 mL of the following 1:1 mixture of CH_2_Cl_2_: MeOH and CHCl_3_: EtOH.

#### 2.2.3. Surfactant Selection and Polymer Concentration

The composition of aqueous phase was kept constant during the whole optimization method using 5 mg/mL of Pluronic F-127 in acetic acid 0.1%. Tween 80 and Span 80^®^ separately were tested as surfactants. Once selected, 2 mg/mL, and 7 mg/mL of Pluronic F 127 were evaluated.

#### 2.2.4. Preparation, Characterization, and In Vitro Evaluation of DMC and BDM-Loaded PLHNs

A total of 250 mg of Pluronic F-127 were dissolved in 50 mL of acetic acid 0.1% and 2 g of a 1:1 mixture of Tween 80: Span80 1:1 was added. The organic phase was composed of 120 mg of Chol and 5 mg of DMC or BDM dissolved in 6 mL of MeOH: CHCl_3_ 1:1. Then, the aqueous solution was added into the organic one at medium speed and homogenized at 16,000 rpm for 10 min to form an appropriate emulsion. The nanoparticles were collected by ultracentrifugation using a Thermo Scientific Sorvall ST 16R centrifuge (Thermo Fisher Scientific, Waltham, MA, USA) at 12,000 rpm for 40 min at 10 °C. To remove the remaining unencapsulated substrate and unreacted substances, the emulsion was washed three times with ultrapure water. The final formulation was dispersed in 5 mL of purified water containing 0.01% Tween80^®^; filtered through an ADVANTEC^®^ ultrafilter unit and refrigerated up to further characterization. Blanks of PLHNs were prepared as mentioned above without adding DMC or BDM in the organic phase. In Figure 2 is presented the schematic representation of the method procedure.

### 2.3. Characterization Techniques

#### 2.3.1. Encapsulation Efficiency (EE)

The EE was calculated through direct and indirect methods using the Equations (1) and (2), respectively. For the direct method the amount of CUR, BDM, or DMC in the three independent formulations of nanoparticles was determined for each curcuminoid by taking 100 µL of fresh PLHNs dissolved in 900 µL of MeOH. Meanwhile, for the indirect method, the amount of free curcuminoid was determined in the supernatant collected by ultracentrifugation using a Thermo Scientific Sorvall ST 16R centrifuge (Thermo Fisher Scientific, Waltham, MA, USA) at 12,000 rpm for 40 min at 10 °C. The solutions for both the direct and indirect methods were filtered through a 0.45-μm cellulose acetate membrane placed in a Sartorius stainless steel syringe filter holder. A total of 10 μL of the samples were injected in a Dionex Ultimate 3000 UHPLC system (Thermo Fisher Scientific, Waltham, MA, USA) equipped with a variable wavelength detector, pump, variable temperature compartment column and autosampler. The chromatographic elution was carried out in a Nucleosil 100-5 C18 column (250 mm × 4.0 mm, 5 μm) at a temperature of 35 °C using 55% of MeCN and 45% of H_3_PO_4_ 0.1% as mobile phase at a flow rate of 1 mL/min and setting down the detection at 420 nm.
(1)$${\mathrm{EE}}_{D}=\frac{\mathrm{Drug}\;\mathrm{in}\;\mathrm{nanoparticle}\;}{\mathrm{Total}\;\mathrm{drug}\;\mathrm{added}}\times 100\;$$
(2)$${\mathrm{EE}}_{I}=\frac{\mathrm{Total}\;\mathrm{drug}\;\mathrm{content}\;\left(\mathrm{mg}\right)-\mathrm{free}\;\mathrm{drug}\left(\mathrm{mg}\right)\;}{\mathrm{Total}\;\mathrm{drug}\;\mathrm{content}\;\left(\mathrm{mg}\right)}\times 100\;$$

#### 2.3.2. Fourier Transform Infrared (FT-IR)

The FT-IR spectra of the sample were recorded on a Thermo Scientific Nicolet 6700 FT-IR spectrometer (Thermo Fisher Scientific, Waltham, MA, USA) fitted with a diamond attenuated total reflectance (ATR) accessory. The samples were placed directly into the ATR cell without further preparation and analyzed in the range of 4000−600 cm^−1^.

#### 2.3.3. Dynamic Light Scattering (DLS)

The particle size (z-average) and polydispersity index (PI) were measured on the basis of the DLS technique on a Malvern Nano Zetasizer ZS90 instrument (Malvern Panalytical, Malvern, UK) using the medium refractive index of 1.33, and viscosity 0.8872 cP under 90°. The samples were diluted with deionized water to achieve the appropriate concentrations, and the measurements were performed at 25 °C.

#### 2.3.4. High Resolution Transmission Electron Microscopy (HR-TEM)

The PLHN morphology was evaluated using a JEOL, JEM2011 HR-TEM (JEOL Ltd., Tokyo, Japan) at an acceleration voltage of 120 kV. The samples were prepared by placing 5 μL of NP suspensions and drying under a nitrogen atmosphere.

#### 2.3.5. Differential Scanning Calorimetry (DSC)

The DSC curves of PLHNs were obtained in a TA Instruments DSC-Q200 calorimeter (TA Instruments, New Castle, DE, USA) equipped with a TA Refrigerated Cooling System 90. Approximately 2 mg of each sample were placed in aluminum pans with lids and the measurement were carried out under a dynamic nitrogen atmosphere of 50 mL/min, a heating rate of 10 °C/min and a temperature range from 40 to 250 °C.

#### 2.3.6. In Vitro Studies

##### Drug Release Profile

The DMC and BDM in vitro release profile from the PLHN as well as the dissolution profile of pure DMC and BDM (used as a reference) were estimated using two different dissolution media, M1 and M2. Briefly, 1 mL of PLHN was immersed in 80 mL of phosphate buffered saline of pH 7.4, containing MeOH 20% and 2.5% of Tween 80 (M1) [37] and water (M2) maintained at 37 ± 0.5 °C and 150 rpm in a Labnet 211 DS shaking incubator (Labnet International Inc., Edison, NJ, USA). Then 4 mL of each solution were withdrawn at specific time intervals without replacing the volume. The aliquots were centrifuged at 6000 rpm for 10 min in a Thermo Scientific Sorvall ST 16R centrifuge at 37 °C. The concentration of DMC and BDM in the solutions were measured using a Shimadzu 1800 double beam UV-Vis spectrophotometer (Shimadzu Corporation, Tokyo, Japan) at a wavelength of 420 nm. The sampling was performed in triplicate.

##### DPPH Radical-Scavenging Activity

The DPPH evaluation was performed as previously reported [38], for both free and nanoencapsulated curcuminoids DMC and BDM. The free curcuminoid samples were evaluated in an ethanolic and aqueous solution. This last medium was also used for curcuminoid nanoparticles samples. In addition, Trolox was used as a standard [39] to assess the DPPH method applied. Briefly, a solution of 2,2-diphenyl-1-picrylhidrazyl (DPPH, 0.25 mM) was prepared using EtOH as a solvent. Next, 0.5 mL of this solution were mixed with Trolox or the respective free or nanoencapsulated curcuminoid solution at different concentrations and incubated at 25 °C in the dark for 30 min. The DPPH absorbance was measured at 517 nm using a Thermo Scientific Genesys S10 spectrophotometer (Thermo Fisher Scientific, Waltham, MA, USA). The controls were prepared for each assay using solvent instead of Trolox or samples. The percentage of the radical-scavenging activity inhibition was calculated for each concentration according to Equation (3). This percentage was plotted against the Trolox or samples concentrations to calculate IC_50_, which is the amount required to reach 50% inhibition of DPPH radical-scavenging activity. The Trolox and samples were analyzed this way in three independent assays.
(3)$$\mathrm{Inhibition}\;\mathrm{percentage}=\frac{\left(\mathrm{Absorbance}\;\mathrm{of}\;\mathrm{control}\;-\;\mathrm{Absorbance}\;\mathrm{of}\;\mathrm{Trolox}\;\mathrm{or}\;\mathrm{sample}\right)}{\left(\mathrm{Absorbance}\;\mathrm{of}\;\mathrm{control}\right)}\times 100$$

## 3. Results

### 3.1. Method Design and Optimization Using CUR as Model

The reported techniques to obtain PLHNs are classified into one-step and two-step approaches. The one-step synthesis is based on the nanoprecipitation and self-assembly technique [40]. According to Zhang et al. (2010), the water-miscible organic phase containing both the polymer and hydrophobic drug is added dropwise into the aqueous phase composed of the lipid and a small quantity of a water-miscible organic solvent for facilitating lipid dissolution [40]. The method proposed herein was also based on a one-step approach. However, in contrast to the existing techniques, it consisted of incorporating the aqueous phase containing the polymer into the organic one composed of the lipid and CUR, taking advantage of the components’ solubilities in the respective phases. Therefore, reducing the necessity of including organic solvent in the water solution.

Designing and optimizing methods for nanoparticle preparation using traditional statistical analysis is a complex and time-consuming process that requires plenty of experiments [41]. In addition, a complete characterization of each prepared formulation is quite expensive because it requires advanced instrumentation that may not be available in every laboratory. Therefore, as an alternative, the strategy developed in this contribution was to perform a screening of variables considering first the emulsion formation which can be visually assessed [42], and a second step determining EE_I_ and EE_D_. In this regard, high EE values are desirable for the delivery of higher amounts of drug payload. Consequently, in terms of the scalability and industrial development of these materials, high EE implies the economical usage of drugs without a decrease in their therapeutic index [41,43]. Further, EE can be influenced by the method used to carry out the encapsulation process, the partition coefficient of the target molecule in the solvents used and the size distribution of the PLHNs [43,44]. All the conditions and parameters evaluated are shown in Table 1.

#### 3.1.1. Homogenization Technique

The structural organization of the PLHNs and the high EE depends directly on the preparation of the nanoemulsion, and the drug-polymer interactions [15]. Mixing the organic and aqueous together and applying the ultrasonication probe, neither for 30 nor 45 min, provided a visual emulsion. In contrast, using high-speed homogenization, adding aqueous solution dropwise into the organic one, and keeping the homogenization speed at 10,000 rpm for 10 min after the addition resulted in a successful emulsion with an EE of 92%. Then, instead of using the fast addition of the aqueous phase into the organic one, the medium and fast adding speeds were investigated. The optimum addition speed showed to be the medium one resulting in 98% of EE against 30% and 46% of the fast and slow speeds, respectively. Previous reports [14,30] indicate the importance of the rate of precipitation of the hydrophobic drug and the polymer, for instance, with similar or equal rates of precipitation of the two species, the homogeneous particles would be obtained while large differences between rates will force the selective precipitation of each component affecting the encapsulation of the drug [30]. Further, the tested 12,000 and 16,000 rpm homogenization speed resulted in 92% and 98% of EE respectively. Consequently, these last conditions were selected to continue the method optimization.

#### 3.1.2. Lipid, Drug Loading, Lipid-Drug Ratio, and Organic Solvent

Instead of Chol, four lipids cocoa butter, cetyl palmitate, stearic acid, and a 1:1 mixture of cetyl palmitate:Chol were incorporated into the organic phase and evaluated for emulsion formation and EE. All the formulations provided emulsions and an acceptable EE; stearic acid was the best with 80%. However, the formulation containing Chol continued to exhibit the best EE.

In an attempt to achieve a high drug loading capacity into the PLHNs, increments in the amount of CUR were investigated by adding 7.5, 10, and 15 mg to the organic phase in individual experiments. Increases in drug content decreased the EE until it was not forming an emulsion when using 15 mg of CUR. Further, two additional lipid-drug ratios using a lower 12:1 and a higher 48:1 lipid quantity in relation with the CUR content resulted in 75 and 70% EE, respectively, which did not represent an improvement in comparison with the 24:1 selected as starting parameters.

Finally, two different 1:1 solvent mixtures were tested as an organic phase, CH_2_Cl_2_: MeOH and CH_3_Cl_3_: EtOH. However, the results showed a deficient EE with both solvent mixtures yielding 37% and 8% respectively.

#### 3.1.3. Surfactant Selection and Polymer Concentration

The starting parameters in the development and optimization of this method included a 1:1 mixture of Tween 80: Span 80^®^. It was reported that attraction between both Tween 80 and Span 80 surfactants may affect the loading [45]. Therefore, Tween 80 and Span 80^®^ were separately tested in the aqueous solution in order investigate the effect of using both surfactants together in the formulation. Results indicated that the use of only one of them separately did not provide an emulsion; therefore, both surfactants are needed in the formulation to successfully prepare the PLHNs. Optimizing the polymer concentration by decreasing to 2 mg/mL and increasing up to 7 mg/mL of Pluronic F-127 resulted in a considerable decline in the EE resulting in 5% for 2 mg/mL and in 16% for the 7 mg/L polymer concentration.

In sum, the parameters tested provided consequently several formulations in which acceptable EE values were observed when emulsions were successfully obtained. This can be attributed to the important role of Pluronic in the shell-core incorporation providing the forces to assemble via the attraction between alkyl groups in the polymer and aromatic groups in CUR [46]. The symmetric electron density distribution of atoms for the CUR in the Pluronic’s non-polar part can lead to the assembly of electrostatic van der Waals forces. In the hydrophobic PO chain of Pluronic, there is only one –CH_3_ per monomer while Chol can interact with the hydrophobic part of the chain [47]. This may lead to more interaction sites for CUR with the polymer, thus leading to higher loading into PLHNs.

The encapsulation was confirmed by FT-IR measurements in formulations exhibiting high EE values. FT-IR was performed to evaluate the successful incorporation of CUR loaded into PLHNs. Figure 3 shows the FT-IR spectra of the Chol, Pluronic F-127, plain PLHNs, and CUR-loaded PLHNs. The main peaks for the CUR-loaded PLHNs correspond to 3363 cm^−1^ related to the stretching vibration of hydrogen-bonded (-OH), 1649 cm^−1^ of the C=O stretching vibration, and 1270 cm^−1^ due to C-O stretching. The CUR-loaded PLHNs spectra indicated a combination of these signals from CUR as well as from Pluronic F-127 and Chol components. For instance, bands at 2897 cm^−1^ associated with C-H stretch aliphatic and at 1359 cm^−1^ corresponding to in-plane O-H bend pertain to Pluronic F-127 [48]. In addition, signals at 1449 cm^−1^, 1043 cm^−1^ and 742 cm^−1^ correspond to CH_2_ and CH_3_ deformation vibrations, ring deformation, and C-H out-of-plane bending from Chol [49]. The fact that some other signals reported for CUR [50,51] are not predominant and the combination of CUR main bands with Pluronic F-127 and Chol signals in CUR-loaded PLHNs spectra confirm that CUR was successfully loaded into PLHNs.

### 3.2. Preparation, Characterization and In Vitro Evaluation of DMC and BDM-Loaded PLHNs

#### 3.2.1. Encapsulation, FT-IR and EE

We then confirmed the encapsulation FT-IR measurements were performed in the nanoformulations. Comparing the FT-IR spectra of plain PLHNs, DMC, or BDM-loaded PLHNs with the ones obtained for pure DMC or BDM (Figure 4), it becomes evident that they share peaks at wavenumbers of 3318 cm^−1^ associated with stretching vibration of hydrogen-bonded (-OH), as well as at 1659 and 1707 cm^−1^ stretching vibration of conjugated carbonyl (C=O) group, 1459 cm^−1^ and 1422 cm^−1^ CH- bending, and 1370 cm^−1^ and 1347 cm^−1^ to in-plane O-H bend, respectively, for DMC and BDM. Besides, characteristic bands at 2928 cm^−1^ associated to C-H stretch aliphatic corresponding to Pluronic F-127 [48] and signals at 1048 cm^−1^ and 754 cm^−1^ corresponding to ring deformation and C-H out-of-plane bending from Chol [49] are also present in the curcuminoids loaded PLHNs spectra. Therefore, these facts are to be expected due to the combination of signals of the components and the pure drug, which is further confirm that curcuminoids DMC and BDM were successfully loaded onto PLHNs core.

Concerning the EE of DMC and BDM, both DMC and BDM were efficiently loaded in PLHNs, achieving an encapsulation efficiency of 95% and 92%, respectively. Therefore, a high EE meant that the curcuminoid maximal solubility in the lipid was reached in the PLHNs and that all the molecules remained in the particles after lipid solidification [52]. The high values of EE can be attributed to phenyl groups on the curcuminoids structure loaded into HPLN. In addition, Pluronic has an important role in the shell-core incorporation, because it provides the forces to the assembly via the attraction between alkyl groups in the polymer and aromatic groups [46]. The structural organization of the PLHNs and the high EE depends directly on the preparation of nanoemulsion and the drug-polymer interactions. Previous reports [14,30] indicate the importance of the rate of precipitation of the hydro-phobic drug and the polymer, large differences between rates will force the selective precipitation of each component, consequently affecting the encapsulation of the drug [30]. 

#### 3.2.2. DLS and HR-TEM Techniques

The particle size and size distribution in terms of PDI values were evaluated by DLS whereas the morphology of the prepared PLHNs was observed by HR-TEM. Figure 5 presents the HR-TEM images and histogram of the size distribution of the DMC and BDM-loaded PLHNs. For the characterization of nanosystems for drug delivery, parameters such as average size and polydispersity index (PDI) are considered one of the most important factors to evaluate the stability and the proper function of the nanoparticles due to the influence in the loading and release of the compound inside the nanoparticle [53,54]; therefore, it is important nanoparticles present high reproducibility and homogeneity [53]. In this concern, PDI defines the variation in particle size distribution within the nanoemulsion. According to Souza et al. (2014) and Valencia et al. (2021) PDI values ≤ 0.4 are considered monodisperse, which implies that there is uniformity in the sample size [53,55], and it also indicates that the formulation has a low aggregation of the sample during isolation or analysis [56]. 

The results showed that the DMC loaded nanoparticles exhibited a particle size of 75.55 ± 0.51 nm with a PDI of 0.281 ± 0.014 and the BDM nanoparticles 15.13 ± 0.014 nm with a PDI of 0.196 ± 0.032. A formulation composed of Chol, lecitin and vitamin E TPGS reports a BDM-loaded PLHNs size of 75.98 nm [57] while Dolatabi et al. [58] reported a DMC PLHNs formulation with Precirol^®^ ATO5 and polaxamer 188 with a-size of 160.7 nm. Our findings indicate smaller particle sizes than both studies and a similar average size of (a-size) <50 nm for BDM in a PLHNs formulation with ethyl oleate and PEG-400 [59]. Due to the low size, the PLHNs can be absorbed by systemic circulation in the intestine [60] which improves the bioavailability of these small molecules.

In respect to morphology, the TEM images of DMC and BDM-loaded PLHNs present nanoparticles with a spherical appearance. These findings are in agreement with results from the literature reporting a similar shape for other curcuminoid polymeric nanoparticles [59].

#### 3.2.3. Differential Scanning Calorimetry (DSC)

The thermal evaluation can reveal the solid state of the encapsulated drug, it provides information about the microcrystalline form and if present any polymorph change or transition change in amorphous form [61]. In addition, DSC can show any incompatibility or possible interaction between the drug and excipients, which may affect the efficacy of the encapsulated drug [62]. The DSC curves of formulation components are presented in Figure 6. DMC and BDM exhibited a unique endothermic event at 173 °C and 238 °C, respectively, in agreement with values reported in literature [63]. Further, Pluronic F-127 and Chol were observed at 58 °C and 147 °C, respectively, corresponding to their melting temperatures and in the case of Chol to its monohydrate form [64]. The DSC curves of DMC and BDM-loaded PLHNs presented similar thermal behavior. One broad endothermic event between 40 and 70 °C is coincident with the melting temperature of Pluronic F-127 and water loss. Then, a sharp endothermic one around 100 °C associated to a solvated form of Chol [65] that can be crystallized during PLHNs preparation. The endothermic event related to free curcuminoids was not observed in the nanoparticles which is an indicative that they were molecularly dispersed within the PLHNs matrix [66,67]. Moreover, there was no evidence of incompatibilities between curcuminoids and formulation constituents.

#### 3.2.4. In Vitro Studies

##### Drug Release

The in vitro release profiles of DMC and BDM from PLHNs as well as the dissolution profiles of free curcuminoids in two dissolution media are shown in Figure 7. Medium 1 (M1) composed of phosphate buffered pH 6.8 with 20% of MeOH and medium 2 (M2) was water. Several media and pH values were tested with anomalous results due to the chemical instability of curcuminoids at pHs lower than 7. Its degradation compromised the detection and quantification of curcuminoids [68,69]. In this regard, reports in the literature have used pH below 7 in the release of curcuminoids [70,71,72]. In addition, the absorption pH in the lumen of the intestine has been reported to be in the range of 6.8–7.4; therefore, was considered an appropriate media for curcuminoids [73]. In respect to the use of water as a medium, it was evaluated considering the main purpose of this study on preparing nanoencapsulated curcuminoids to improve their water solubility; furthermore, in light of some studies that have reported the degradation of the Pluronic backbone in water which can contribute to water solubilization and subsequent release of the drug [4,5,6].

The presence of MeOH in M1 suggested an increased dissolution rate of free curcuminoids and a higher release profile of curcuminoids from the PLHNs. However, release and dissolution profiles obtained in both media were not significantly different from BDM. Nevertheless, comparing the dissolution profile of free DMC and BDM with curcuminoids released from PLHNs in water, showed promising results. At 180 min only 0.1% of the free DMC was dissolved while its release from the hybrid nanosystem was 88% at 180 min. In turn, only 0.3% of free BDM dissolved after 180 min while 68% were released from the nanoparticle at 180 min. The difference in values in favor of DMC can be explained due to the different pka values (DMC < BDM), which can destabilize the keto-enol structure and consequently the dissociation of the enol hydrogen affecting the stability and the solubility of the curcuminoids [74]. Overall, these results suggested that this formulation would significantly improve the bioavailability of these curcuminoids.

The release of a loaded drug molecule from the shell-core largely depends on hydrophobic interactions between the inner core and drug, as previously mentioned. The increased release of DMC and BDM from the PLHNs can be attributed to the hydrophobic interaction between the curcuminoid and the bilayer, as the hydrophobic interaction becomes weak, the shell core breaks and exhibits a fast and sustained release of the molecule [75].

##### Antioxidant Activity Evaluation of DMC and BDMC Free and Loaded into PLHNs

The antioxidant activity of Trolox, free and PLHN-loaded DMC and BDM was studied through a DPPH analysis, as described in the Materials and Methods section. Trolox was used as a standard to assess the DPPH method, obtaining adequate results (R^2^ = 0.9956) that allowed to determine an IC_50_ of 5.62 µg/mL. Further, results of the samples antioxidant activity are shown in Table 2.

Evaluation of the antioxidant activity of the free curcuminoid dissolutions in EtOH indicated DMC (R^2^ = 0.9979) yielded the lowest IC_50_ representing an antioxidant activity 28% higher than the antioxidant activity of BDM (R^2^ = 0.9930). This observation is consistent with the trend previously reported for the antioxidant activity of the main curcuminoids, where DMC was found to yield higher antioxidant values than BDM [76]. A DPPH analyses of the aqueous solutions of the free curcuminoids showed a considerably higher IC_50_ for both samples; hence, much lower antioxidant activity, which is associated with their low solubility in water. Results showed an opposite trend of antioxidant activity in these aqueous samples, with BDM (R^2^ = 0.9947) yielding the lowest IC50, indicating an antioxidant activity 35% higher than the antioxidant activity of DMC (R^2^ = 0.9932) in water, which is in alignment with BDM relative higher aqueous solubility in respect to DMC [77].

DPPH assays of DMC and BDM-loaded PLHNs showed an important decrease in the IC_50_, therefore a much higher 80-to 160-fold antioxidant activity than free curcuminoids in aqueous solution. As shown by the results from a one-way analysis of variance (ANOVA) (Table 1), both the DMC (R^2^ = 0.9936) and BDM (R^2^ = 0.9960)-loaded PLHNs presented similar antioxidant activity to the one obtained for the ethanolic solution of the free curcuminoids. Further, BDM nanoparticles showed a lower IC_50_ than free BDM ethanolic solution, with BDM-loaded PLHNs antioxidant activity improving by 11%, aligning with results using CUR that showed antioxidant activity improved by 17% [78]. In sum, our results show that the antioxidant activity of DMC and BDM was significantly enhanced by the PLHNs preparation, consistent with previous results on polymeric nanoparticle formulations of curcuminoid mixtures [79] and the individual curcuminoids DMC and BDM [59], aligning also with results obtained for other polyphenols, such as hesperetin formulations [80].

## 4. Conclusions

DMC and BDM-loaded polymer-lipid hybrid nanoparticles (PLHNs) composed of Pluronic F-127 and cholesterol were synthesized and characterized through a simple method designed and optimized using CUR as a model curcuminoid. The influence of the encapsulation efficiency of various processing parameters including the ultrasonic probe, high-speed homogenization, mixing phases and homogenization speed were evaluated. Organic and aqueous phases composition: lipid, drug loading, lipid-drug ratio, organic solvent, surfactant, and polymer concentration was systematically assessed. The study confirmed that Pluronic F-127 plays an important role in the shell-core incorporation providing the forces to assemble via the attraction between alkyl groups in the polymer and aromatic groups in curcuminoids. Therefore, contributing to improving the solubility, stability, and manages to successfully encapsulate the curcuminoids by the PLHNs, as well and maintain a high loading efficiency as previously reported in the literature. The method for PLHNs synthesis developed and optimized herein presents advantages in terms of preparation, appropriate nanoparticles characteristics including particle size, monodisperse size distribution, shape, and encapsulation efficiency. Further, the DMC and BDM-loaded PLHNs exhibited considerable improvements in aqueous dissolution consequently in their antioxidant activity. Therefore, this method can be extended to other curcuminoids and molecules with a similar backbone structure to improve the bioactivities associated with their limited bioavailability.

## Figures and Tables

**Figure 1 polymers-13-04207-f001:**
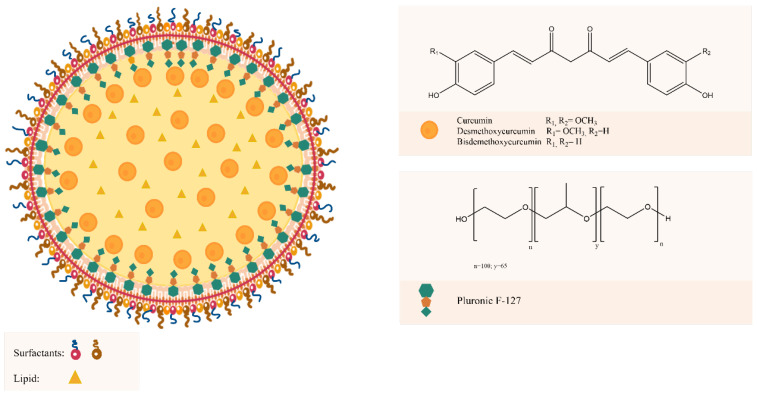
Schematic representation of PLHNs and chemical structure of Pluronic F-127, CUR, DMC, and BDM.

**Figure 2 polymers-13-04207-f002:**
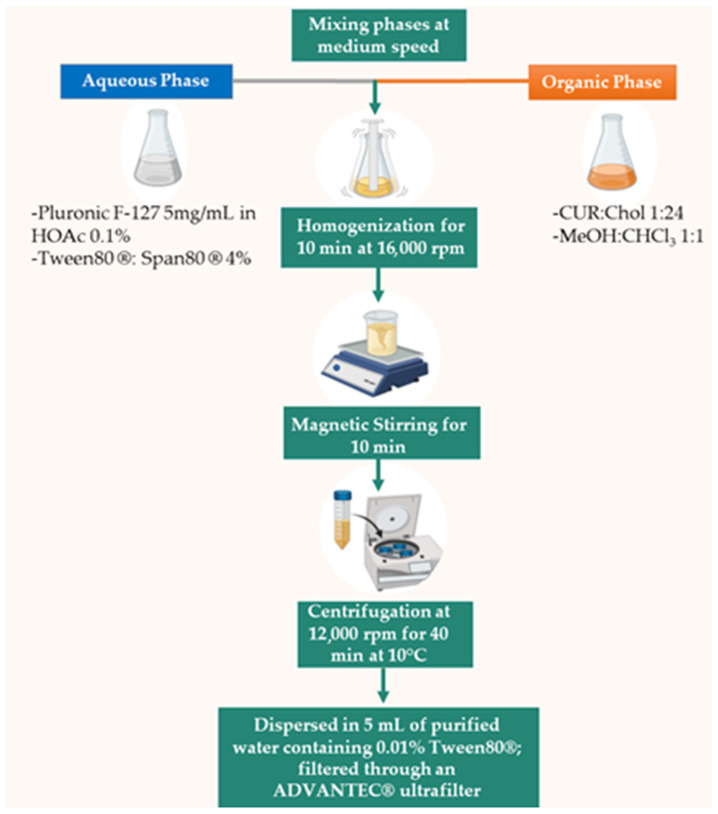
Schematic representation of the developed and optimized method procedure.

**Figure 3 polymers-13-04207-f003:**
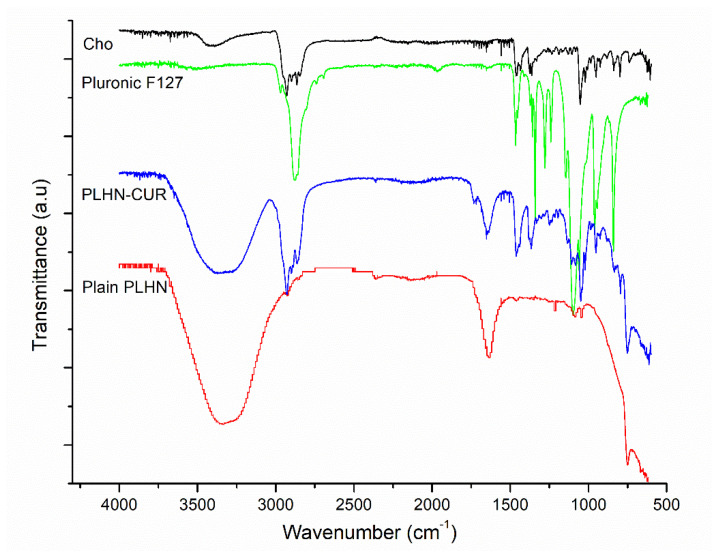
FT–IR spectra of PLHNs main components, CUR–loaded PLHNs and plain PLHNs.

**Figure 4 polymers-13-04207-f004:**
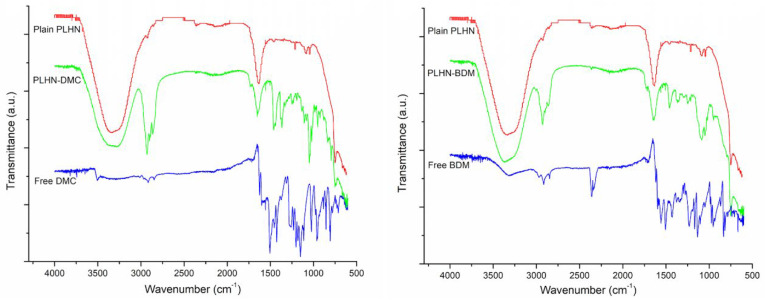
FT–IR spectra of DMC and BDM–loaded PLHNs, plain PLHNs and free DMC and BDM.

**Figure 5 polymers-13-04207-f005:**
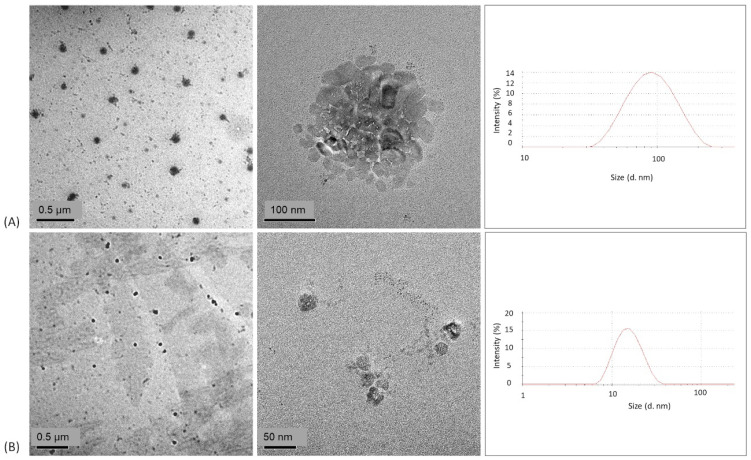
TEM images and size distribution histogram of (**A**) DMC and (**B**) BDM–loaded PLHNs.

**Figure 6 polymers-13-04207-f006:**
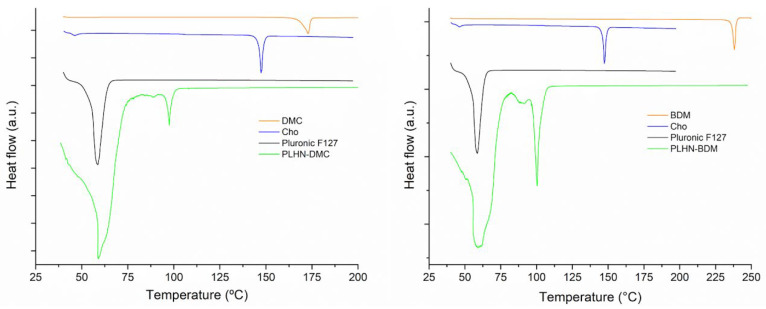
DSC curves of PLHNs main components and the DMC and BDM–loaded formulation.

**Figure 7 polymers-13-04207-f007:**
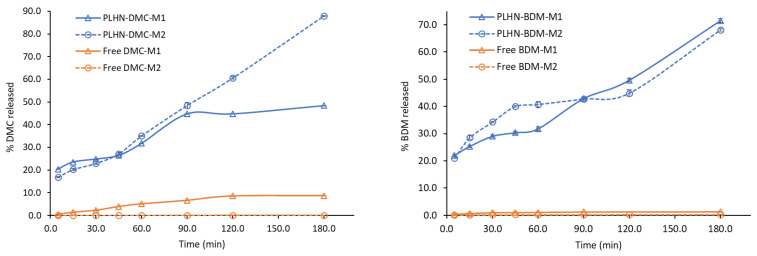
Release profile of DMC and BDM from the PLHNs compared with free DMC and BDM dissolution rate in two dissolution media. Error bars represent the standard deviation of DMC and BDM concentration in the triplicates.

**Table 1 polymers-13-04207-t001:** Parameters and conditions tested in the development and optimization of the method.

1-Homogenization Technique ^1^			N° Formulation	Emulsion Observed	EE (%)
Ultrasonic Processor One-time addition, 130 watts, 20 kHertz, 3 mm Ti probe	Sonication Time	30 min	1	No	NA
		45 min	2	No	NA
High Speed Homogenizer 10,000 rpm, 10 min	Mixing phases speed	Slow	3	Yes	46
		Medium	4	Yes	98
		Fast	5	Yes	30
**2-Optimization of homogenization technique ^1^**				**Emulsion observed**	**EE (%)**
High speed homogenizer, mixing phases at medium speed, 10 min	Homogenization speed	12,000 rpm	6	Yes	92
		16,000 rpm	7	Yes	98
**3-Lipid, drug loading, lipid-drug ratio, and organic solvent**				**Emulsion observed**	**EE (%)**
Type of lipid	24:1 lipid:drug ratio	Cocoa butter	8	Yes	71
		Cetyl Palmitate	9	Yes	54
		Stearic acid	10	Yes	81
		Cholesterol: Cetyl Palmitate 1:1	11	Yes	47
Drug loading	mg of CUR	7.5	12	Yes	87
		10	13	Yes	62
		15	14	No	NA
Lipid-drug ratio	Chol:CUR	12:1	15	Yes	75
		48:1	16	Yes	70
Organic solvent	1:1 solvent mixture	CH_2_Cl_2_:MeOH	17	Yes	37
		CH_3_Cl_3_:EtOH	18	Yes	8
**4-Surfactant selection and polymer concentration**				**Emulsion observed**	**EE (%)**
Surfactant selection	1% surfactant concentration	Tween 80	19	No	NA
		Span 80	20	No	NA
Polymer concentration	mg/mL of Pluronic F-127	2	21	Yes	5
		7	22	Yes	16

NA: Non-applicable; ^1^ Starting composition: aqueous phase containing 5 mg/mL of Pluronic F-127 in acetic acid 0.1% and Tween 80: Span80 1:1 4% and organic phase containing a CUR-lipid ratio of 1:24 in a 1:1 mixture of MeOH: CHCl_3._

**Table 2 polymers-13-04207-t002:** Antioxidant activity of free and curcuminoids DMC and BDM-loaded PLHNs.

IC_50_ (µg/mL) ^1,2,3^			
	EtOH	Water	PLHN
DMC	12.46 ^a,#^ ± 0.02	2143.07 ^a,&^ ± 0.61	12.74 ^a,#^ ± 0.09
BDM	17.94 ^b,^^ ± 0.06	1398.68 ^b,*^ ± 5.07	16.03 ^b,^^ ± 0.55

^1^ IC_50_ µg/mL for each curcuminoid. ^2^ Values are expressed as mean ± standard deviation (S.D.). ^3^ Different superscript letters in the same column or different superscript signs in the same row indicate differences are significant (*p* < 0.05) using one-way analysis of variance (ANOVA) with a Tukey post hoc.

## Data Availability

Not applicable.

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
