# Peer review of "Design of Hybrid Polymeric-Lipid Nanoparticles Using Curcumin as a Model: Preparation, Characterization, and In Vitro Evaluation of Demethoxycurcumin and Bisdemethoxycurcumin-Loaded Nanoparticles"

_polymers, 2021, doi:10.3390/polym13234207_

Round 1

Reviewer 1 Report

The Authors describe interesting manuscript, but some elements need to be improved:

1. Could the Authors provide tables with the composition of the formulation. Iit is unclear how many formulations were made and which exactly were tested and why?

2. In the text  appears: “error! reference source not found”

3. The manufacturer and country should be added to the names of the equipment

4. In the Section 2.3.1. – EEd – its percent loading? What means EE1?

5. Why 80 ml of medium was used to study the release profile? why water was used?

6. Where are the EE data? maybe a table should be made

7. In the section 3.2.1. Autohors should describe EE but not FT-IR

8. Standard deviation and statistical data are missing in the data gel (e.g. drug release plot, data in the text)

9. Have nanoparticles formulations been studied in the dsc and FT-IR research?

10. Where is information about methods of TEM images (or SEM)

Author Response

Revisor 1

  1. Could the Authors provide tables with the composition of the formulation, Iit is unclear how many formulations were made and which exactly were tested and why?

A table has been included in the manuscript as Table 1 and a scheme representing the process has been included as Figure 2.

  1. In the text  appears: “error! reference source not found”

The error has been corrected.

  1. The manufacturer and country should be added to the names of the equipment

The manufacturer and country of equipments have been added

  1. In the Section 2.3.1. – EE – its percent loading? What means EE1?

Yes, in effect encapsulation efficiency (EE%) is an indicator of drug loading efficiency. EEI corresponds to the method in which the unentrapped drug is determined in the supernatant after centrifugation. This has been clarified in the text.

  1. Why 80 ml of medium was used to study the release profile? why water was used?

The volume was selected according to the previously reported methods for curcumin which indeed used 20mL (Mater. Sci. Eng. C 2020, 109, 110576, doi:10.1016/j.msec.2019.110576). Further, it was also considered the following aspects: 1) the poorly water solubility of curcumin, 2) ensuring critical micelle concentration during release profile and 3) having a concentration of curcumin in the aliquots enough to be detected and quantified by the analytical equipment.

In respect to the use of water as a medium, it has been justified in the manuscript in the Section 3.2.4 and some references have been added.

  1. Where are the EE data? maybe a table should be made

A table has been prepared and included in the manuscript as Table 1

  1. In the section 3.2.1. Autohors should describe EE but not FT-IR

In the section “3.2.1 Encapsulation and encapsulation efficiency” both FT-IR and EE need to be described as FT-IR is an appropriate technique to confirm encapsulation. However, attending to your comment, the description related to FT-IR and EE has been separated in the text and further discussion about EE was added.

  1. Standard deviation and statistical data are missing in the data gel (e.g. drug release plot, data in the text)

The drug release profile was done in triplicate, the error bars were included in the plot. However, as the plotted y-axis is the percentage the error bars are quite small.

  1. Have nanoparticles formulations been studied in the dsc and FT-IR research?

Yes, nanoparticles formulations were studied by both techniques. Sections 2.3.2 and 2.3.5 in Material and methods as well as in sections 3.2.1 and 3.2.3 in the Results.

  1. Where is information about methods of TEM images (or SEM)

TEM method information is in section 2.3.4

Reviewer 2 Report

In the present research work, authors have formulated hybrid polymeric-lipid nanoparticles. This article is well within the scope of “Polymers”. Although the work is good, but author’s need to address the following comments or suggestions:

  1. It is not very clear that you have prepared nanoparticles are prepared only for one or all (from curcumin, demethoxycurcumin and bisdemethoxycurcumin).
  2. Entrapment efficiency procedure is not clear. Refer the formula given in “Brazilian J Pharm Sci. 2020;56:e18069”.
  3. What was the ultracentrifugation speed adjusted for entrapment efficiency study?
  4. Are you sure that you have recorded the Brownian movement of particles at 90º angle in DLS?
  5. Why did you do the drug release studies at pH 7.4? and why did only at one pH?
  6. In anti-oxidant assay, why standard antioxidant is not taken? I advise you to refer “J Drug Delivery Sci Technol. 2021;66:102873” for the calculation and experimentation.

Author Response

Revisor 2

  1. It is not very clear that you have prepared nanoparticles are prepared only for one or all (from curcumin, demethoxycurcumin and bisdemethoxycurcumin).

       It was clarified in the manuscript.

  1. Entrapment efficiency procedure is not clear. Refer the formula given in “Brazilian J Pharm Sci. 2020;56:e18069”.

       Entrapment efficiency procedure has been clarified in the experimental section 2.3.1 and the formula was slightly modified according to the suggested reference.

  1. What was the ultracentrifugation speed adjusted for entrapment efficiency study?

       Yes, it was 12000 rpm. This information has been included in the section 2.3.1

  1. Are you sure that you have recorded the Brownian movement of particles at 90º angle in DLS?

       Yes, the optics of the equipment used ZETASIZER NANO ZS90 is 90°.

  1. Why did you do the drug release studies at pH 7.4? and why did only at one pH?

       Several media and pH values were tested with anomalous results due to the chemical instability of curcuminoids at other pHs. The justification of the use of pH and only one has been included in the manuscript.

  1. In anti-oxidant assay, why standard antioxidant is not taken? I advise you to refer “J Drug Delivery Sci Technol. 2021;66:102873” for the calculation and experimentation.

       We used Trolox as standard in our antioxidant assays as we had described in previous publication (we added citation). As suggested, we added the standard description in the Materials and Methods section (2.3.6) and in the Results and Discussion section (3.2.4).

       Our calculations had been performed applying the suggested formula for each concentration (based in previous work cited in the paper) and our experimentation included three replicates i.e. three curves with different concentrations (each calculated as per the formula) for each sample. As suggested, we have clarified this further in the Materials and Methods (2.3.6.) & Results and Discussion (3.2.4) sections, adding in this last section the above-suggested reference on hesperetin.

Round 2

Reviewer 1 Report

I'm accepting the manuscript for publication

Reviewer 2 Report

Manuscript can be considered for publication.